# Review of Hybrid Materials Based on Polyhydroxyalkanoates for Tissue Engineering Applications

**DOI:** 10.3390/polym13111738

**Published:** 2021-05-26

**Authors:** Artyom Pryadko, Maria A. Surmeneva, Roman A. Surmenev

**Affiliations:** Physical Materials Science and Composite Materials Centre, Research School of Chemistry & Applied Biomedical Sciences, National Research Tomsk Polytechnic University, 30 Lenina Avenue, Tomsk 634050, Russia; vilajer@gmail.com (A.P.); surmenevamaria@mail.ru (M.A.S.)

**Keywords:** polyhydroxyalkanoates, biopolymers, biocompatibility, biodegradibility, composites, modification

## Abstract

This review is focused on hybrid polyhydroxyalkanoate-based (PHA) biomaterials with improved physico-mechanical, chemical, and piezoelectric properties and controlled biodegradation rate for applications in bone, cartilage, nerve and skin tissue engineering. PHAs are polyesters produced by a wide range of bacteria under unbalanced growth conditions. They are biodegradable, biocompatible, and piezoelectric polymers, which make them very attractive biomaterials for various biomedical applications. As naturally derived materials, PHAs have been used for multiple cell and tissue engineering applications; however, their widespread biomedical applications are limited due to their lack of toughness, elasticity, hydrophilicity and bioactivity. The chemical structure of PHAs allows them to combine with other polymers or inorganic materials to form hybrid composites with improved structural and functional properties. Their type (films, fibers, and 3D printed scaffolds) and properties can be tailored with fabrication methods and materials used as fillers. Here, we are aiming to fill in a gap in literature, revealing an up-to-date overview of ongoing research strategies that make use of PHAs as versatile and prospective biomaterials. In this work, a systematic and detailed review of works investigating PHA-based hybrid materials with tailored properties and performance for use in tissue engineering applications is carried out. A literature survey revealed that PHA-based composites have better performance for use in tissue regeneration applications than pure PHA.

## 1. Introduction

Polyhydroxyalkanoates (PHAs) constitute a family of biopolyesters that are synthesized and accumulate within the cellular structure of prokaryotic cells by bacteria, and they act as carbon and energy reserve materials under conditions of limited nutrient, such as nitrogen, oxygen, phosphorous or magnesium. The carbon source is the main influencing factor for the PHA production at industrial scale, because it affects the cell growth, productivity, molecular mass, quality, and composition of a polymer [1,2]. There are published works, including review papers, addressing utilization of various types of carbon sources, such as whey, waste plant oils, waste animal fats, starch, wheat and rice bran, molasses, wastewater as a cheap carbon sources for PHA production [3,4,5,6].

PHAs are biodegradable, biocompatible, piezoelectric, and thermoplastic and show good barrier properties and controllable thermal and mechanical properties depending on the polymer composition [7,8].

Biocompatibility is the ability of a material to perform a desired function without causing any local or systemic adverse responses in the recipient of the material. The biological rejection of an implant leads to an inflammatory response mediated by immune cells, and it may require the removal of the implant. Although PHA has good biocompatibility [9,10], many authors have shown that hybrid materials based on PHA have greater biocompatibility with different cells [11,12,13,14,15,16,17].

The thermoplasticity, barrier properties and degradability characteristics of PHAs indicate that they can be recycled, which makes them very attractive for use as bioplastics for packaging purposes; however, the high cost of PHAs limits their use as a “green plastic”. In spite of this, interest in PHAs as bioplastics for fighting plastic pollution challenge continues to grow worldwide [18]. PHA production increased from 5.3 million tons to 17 million tons within the year 2013 to 2020 [19]. PHA market size is estimated to be USD 62 million in 2020 and is projected to reach USD 121 million by 2025, at a CAGR of 14.2% between 2020 and 2025 [20]. The market is mainly driven by the rising demand for PHA industries such as food and packaging services, agriculture, biomedical, and some others. Factors such as consumer awareness about the toxicity of the petroleum based and sustainable ecofriendly bioplastics will drive the PHA market. There are key markets for PHA, which are Europe, followed by North America and Asia, in terms of value and volume. With an objective to reduce the total cost of PHAs production, new approaches of utilization of different cheap and eco-friendly carbon sources are employed [21,22,23].

The nontoxicity, biodegradability and biocompatibility characteristics of PHAs suggest their potential uses in the biomedical field, especially in tissue engineering and as implants. Gradual biodegradation of PHA-based scaffolds creates a structure for the formation of new tissue and promotes cell growth; moreover, a second surgery is not required to remove the implant [9]. PHA applications include cardiovascular tissue engineering, bone tissue engineering, nerve tissue engineering, and drug delivery systems.

Only a few members of the PHA family are commercially available and produced on a large scale, including poly(3-hydroxybutyrate) (PHB), poly(3-hydroxybutyrate-co-3-hydroxyvalerate) (PHBV) and poly(3-hydroxybutyrate-co-3-hydroxyhexanoate) (PHBHHx). PHB is the most investigated member of the PHA family. PHB is piezoelectric, crystalline, water insoluble and relatively resistant to hydrolytic degradation; however, it reveals poor mechanical properties and is a highly brittle and stiff material [10]. The copolymer PHBV has better mechanical properties than PHB and is tougher, less stiff, and more flexible. PHBV exhibits both a lower crystallinity and melting temperature and increased elongation to break. PHBV copolymer does not cause inflammatory reactions when implanted in mice and rats [24]. PHBVs have also been shown to support in vitro osteogenesis, which makes them suitable for bone regeneration [25].

PHBHHx is another member of the PHA family with improved mechanical properties compared with both PHB and PHBV. PHBHHx promotes enhanced osteogenic differentiation of mesenchymal stem cells (MSCs) [26] and possesses good biocompatibility with fibroblasts, chondrocytes, nerve cells and osteoblasts compared with polylactic acid (PLA), PHB and PHBV [27].

PHB and PHBV are nontoxic because their degradation products are water, carbon dioxide and D-3-hydroxybutyric acid, which are natural constituents of human blood, and PHA-based biomaterials cause less-severe inflammatory reactions compared to other biopolymers, such as PLAs [28]. D-3-Hydroxybutyric acid increases calcium influx in cultured cells and suppresses their death [29]. Oligo(3-hydroxybutyrate-co-3-hydroxyhexanoate), oligo(3-hydroxybutyrate) and 3-hydroxybutyrate, the main degradation products of PHBHHx, are nontoxic and cause low inflammatory effects [30]. However, the application of PHAs is limited due to their weak mechanical and thermal properties, slow degradation rate, lack of bioactivity, and poor hydrophilic properties. To overcome these disadvantages and improve PHA properties and make it more suitable for biomedical applications, many hybrid PHA-based composites have been investigated [28,31,32,33,34,35]. Several reviews have been published describing production, properties, biocompatibility, and potential applications of pure PHAs [36,37,38,39,40]. Unfortunately, there is a lack of systematic and thorough overview addressing the performance of PHA hybrid materials for tissue engineering and biomedical applications. Thus, this review is focused on the physico-mechanical, chemical, and piezoelectric properties, degradation rate, cellular response of hybrid PHA-based composites and their biomedical applications.

## 2. The Most Important Properties of the Hybrids Based on PHAs

### 2.1. Wettability of the Composites

Hydrophilicity is an important property of scaffolds and defines cell adhesion, proliferation and differentiation in vitro and tissue ingrowth in vivo [41,42,43,44,45]. It has been reported that mammalian cells prefer to adhere and proliferate on the surface with moderate hydrophilicity with a water contact angle in the range of 50–70° [46]. In addition, some serum proteins, such as fibronectin and vitronectin, which are well known to play an important role in cell adhesion, are more susceptible to moderate surface wetting [47]. Hydrophilic surfaces absorb proteins more easily than hydrophobic surfaces, thus making them more suitable for cell spreading and proliferation [48]. The hydrophilicity influences not only the amount and type of serum protein adsorption but also the conformation of these proteins on the surface of the scaffolds, which in turn affects the degree of cell adhesion [49]. Therefore, the improved hydrophilicity could facilitate adsorption of more serum proteins to the surface, which improves cell adhesion. The hydrophobic nature of PHAs limits their applications in the biomedical field. The wettability of the PHA surface can be enhanced by the addition of different fillers as well as surface treatment, thus providing better cell adhesion, spreading and proliferation.

Silk fibroin (SF) is a natural biopolymer used in the human body as a suture material. SF has been employed as a versatile material for tissue-engineered scaffolding due to its biocompatibility and the presence of easily accessible chemical groups for functional modifications. SF have been used to impart hydrophilicity to PHAs. For instance, the water contact angle (WCA) of the PHBHHx/SF electrospun films decreased as the SF content in the blends increased. Human umbilical cord-derived mesenchymal stem cells (hUC-MSCs) showed better adhesion on electrospun PHBHHx/SF (1:1, 1:3) and SF films than on electrospun PHBHHx and PHBHHx/SF (3:1) films [31]. The cell layer was more homogenously widespread and adhered completely onto the electrospun film surface. The addition of SF to PHB decreased the WCA of the PHB/SF nanofibrous scaffold. The PHB/SF composite scaffold (50/50 PHB/SF) showed excellent attachment behavior to L929 and HaCaT cells [50]. Fibroblasts demonstrate better adhesion on PHBV/SF nanofibrous scaffolds than pure PHBV scaffolds since the hydrophilicity of the materials is helpful for the absorption of fibronectin, which is essential for fibroblast adhesion in vitro [51].

Chitosan (CTS) is a natural polymer that is biocompatible and nontoxic with highly availability and low cost and possesses antibacterial activity, and it presents a high mass-loss rate and hydrophilicity [32]. CTS has hydrophilic functional groups on its backbone [43,44] that may increase the hydrophilicity of materials through blending. It has been shown that the addition of CTS to PHB increased the hydrophilicity of PHB and decreased its WCA to ~67° for PHB/20 wt% CTS composites [32]. Contact angle measurements carried out on aligned and random electrospun PHB/CTS revealed that the fibrous scaffolds containing CTS were more hydrophilic than the pure fibers and that the aligned fibers had a lower WCA than random scaffolds. The WCAs of PHB, PHB/15 wt% CTS and PHB/20 wt% CTS are 124°, 62°, 43° for random fibers and 110°, 54°, 43° for aligned electrospun fibers, respectively [52]. Other works have reported that the addition of CTS improves the hydrophilicity of PHAs [53,54,55].

Synthetic hydroxyapatite (HA) is the most widely used bioceramic material and has a similar composition and morphology to the inorganic component of natural bone, which can provide a favorable environment for cell adhesion, osteoconduction and osteoinduction [56]. It has been shown that the addition of mHA to PHB scaffolds decreases the WCA of the composite compared to pure PHB scaffolds [33]. HA deposition on the surface of both nanofibrous and cast flat PHB films turned it hydrophilic [57]. The investigation [58] showed that there are no differences in WCA between pure PHB and PHB/nHA composites. There are some conflicting reports of the effects of HA, including the positive and negative effects, on cell adhesion or proliferation [59,60,61,62,63]; thus, additional trials are required in this field.

The hybrid fibrous PHB/polycaprolactone (PCL) membrane possesses a hydrophilic surface [64]. Modification of PHB/PCL fiber mats with silica decreases the WCA [34]. The addition of graphene oxide (GO) nanosheets significantly enhanced the wettability of the surface of the PHBV biopolymer films [65]. The nanofiber PHB scaffold turned from hydrophobic into hydrophilic in surface characteristic with WCA decreasing from 124° to 44° upon addition of soybean protein nanoparticles (SPN) [66].

The COOH functional groups of carbon nanotubes (CNTs) increase the amount of oxygen on the surface, increasing the quantity of C–O. Thus, the wettability of the PHB scaffolds increases by adding CNTs [16]. In another work, addition of 1 wt% of CNT into electrospun PHB nanofibers decreased WCA by 40° [67]. The PHB/carboxyl multiwalled carbon nanotubes (CMWCNT) composite and PHB–calcium alginate/CMWCNT nanofiber membrane have improved hydrophilicity compared to pure PHB [68,69].

The influence of the nanobioglass (nBG) and microbioglass (mBG) particles on the hydrophilic property of the PHB scaffold was investigated [35]. Hybrid films had significantly increased wettability compared to neat polymer. However, the decrease in WCA was more prominent for the nBG composites than for the mBG composites.

The oxidation of the PHBV membrane in an ozone environment generates polar functional groups such as peroxides, hydroxyl, and carbonyl groups on the surface of the PHBV films. Further grafting of methyl methacrylic acid and covalent immobilization of type I collagen on the surface of PHBV led to WCA reduction and better hydrophilicity [17]. The addition of collagen to nanofibrous PHBV/GO scaffolds with a WCA of 110° made the scaffold hydrophilic with a WCA of 52° [11]. Collagen-coated electrospun PHBV nanofiber films demonstrated better hydrophilic behavior than uncoated films [70].

Plasma surface modification is one of the most promising techniques for enhancing hydrophilicity because it does not alter the bulk properties of the treated material [71,72]. Plasma particles interact with the material and introduce polar functional groups such as hydroxyl, carboxyl and carbonyl groups onto the surface of the substrate depending on the plasma gas. However, hydrophilicity decreases over time due to the “hydrophobic recovery” effect, which is the effect of rearrangement of polar groups towards the bulk of the material to reduce the surface energy [73]. This effect has to be inhibited by covalent immobilization of various bioactive molecules, such as silk, gelatine or collagen [74,75]. The oxygen and nitrogen plasma treatment of PHBV nanofiber mats with and without further immobilization of SF was investigated [76]. Unmodified plasma-treated PHBV mats showed hydrophobic recovery after 14 days. SF-modified nitrogen plasma-treated PHBV mats had stabilized WCA at 70°, while oxygen plasma-treated PHBV presented hydrophobic recovery even after SF immobilization, which is probably due to the repulsion of negatively charged SF with negatively charged oxygen-containing groups on the mat surface. Plasma treatment of SF-coated PHBHHx film improves its hydrophilicity, leading to a larger amount of extracellular matrix (ECM) secretion and better cell migration of human smooth muscle cells [74]. PHB/polyaniline (PANi) electrospun scaffolds surface modification with air plasma for 60 s reduced the water contact angle of the composite from 106° to 29.3° [77]. Collagen cross-linked plasma-modified PHB nanofibrous mats showed a better hydrophilicity of the modified nanofibers compared to the nonmodified mats with an 85° difference in WCA [13]. A scheme of the covalent coupling of the protein with PHB mat fibers is shown in Figure 1.

The hydrophobic character of PHAs can be altered to be hydrophilic with the addition of different materials, as shown in detail above. Hydrophilic surfaces provide attachment of cells to the surface of materials as well as cell spreading and proliferation. The addition of different fillers should not impair other important properties of materials, such as mechanical properties, biocompatibility, and biodegradability, which are necessary for their use in biomedicine. The physico-mechanical properties of the PHA-based hybrids will be discussed in the next section.

### 2.2. Physico-Mechanical Properties

Scaffolds in tissue engineering applications must have sufficient mechanical strength during in vitro culturing to maintain the required space for cell infiltration and formation of ECM. Scaffolds should also provide sufficient temporary mechanical support, matching the mechanical properties of the host tissue as closely as possible, to bear in vivo loading and stress conditions. Thus, scaffolds should be designed with appropriate mechanical properties and degradation rates so that they match the mechanical properties of the injured tissue until the newly grown tissue is remodeled by the host tissue and is able to support in vivo stresses [78,79].

Although the different fillers favorably affect the hydrophobic character of PHA, some of them act simultaneously with an improvement in hydrophilicity and can worsen the mechanical properties of the hybrids. For example, SF-modified PHBHHx films have the maximum tensile strength, and the elongation at break is slightly lower than that of PHBHHx films [12]. The combination of SF with PHB led to an increase in elongation at break for the PHB/SF composite and a decrease in tensile and yield strength in comparison to pure PHB [50]. The increase in CTS in the PHB/CTS composite decreases the tensile strength of the scaffolds [32]. All the PHB/CTS blend scaffolds exhibited lower Young’s moduli than pure PHB scaffolds. These changes in tensile strength and tensile Young’s modulus are related to the lower mechanical properties of CTS in comparison to PHB. The toughness and maximum strain of the scaffolds were enhanced with increasing CTS in the blended scaffolds. This increase is due to the tough nature of CTS in comparison to PHB. In another work, it was reported that the CTS addition in PHB causes a reduction in tensile strength and that the Young’s modulus and tensile strength for aligned PHB/CTS electrospun fibers are greater than those for random fibers [52]. Collagen immobilization decreases the tensile strength of PHBV/collagen electrospun nanofibers [14]. To overcome this disadvantage, the addition of one more filler into the composite may be a prospective option. For example, in the case of the PHBV/GO/collagen composite, collagen did not play any significant role in the mechanical properties of the material [11]. The addition of biphasic calcium phosphate (BCP) particles into the PHB/CTS membrane is beneficial to mechanical properties [80]. Compared to the PHB/CTS and pure PHB membranes, mechanical properties, such as the initial Young’s modulus and ultimate tensile strength, were enhanced after incorporation of BCP. The mechanical properties of the PHB/CTS scaffolds were improved significantly after the addition of multi-walled carbon nanotubes (MWCNTs). MWCNT addition resulted in a significant increase in the scaffold’s elastic Young’s modulus, tensile strength and yield strength [54]. It was found that incorporation of curcumin up to 20 wt% into PHB/MWCNT electrospun scaffolds had a significant effect on increasing ultimate strength values as compared to the neat PHB nanofibers [81].

The addition of mechanically strong materials into the PHA matrices leads to an improvement in the mechanical properties of the composites. The tensile properties of PHBV were significantly increased by the addition of HA nanoparticles (NPs) [24]. The addition of HA to the PHB scaffold allows an increase in the compressive Young’s modulus and compressive strength of the PHB, while the PHBHHx/mHA composite has a decreased compressive elastic Young’s modulus with the same compressive strength compared to the pure PHBHHx scaffold [82]. PHB/10 wt% nHA composite scaffolds have an improved compressive Young’s modulus and compressive strength compared to neat PHB scaffolds [83]. The significant increase in mechanical properties of the composite scaffolds compared to the pure PHB scaffold was due to the homogeneous dispersion of nHA in the matrix. HA NP incorporation within PHB/nHA (blend) fibers significantly improved the mechanical properties of the PHB mats [58]. In contrast, the mechanical properties of the PHB/nHA (spray) framework deteriorated in comparison with those of the neat PHB mat. The tensile strength and strain as well as the elastic Young’s modulus decreased dramatically in the PHB/nHA (spray). PHBV fibers containing 10 wt% nHA or 10 wt% nHA/bredigite (BR) showed higher mechanical strength and Young’s modulus than PHBV fibers incorporated with 10 wt% BR [84]. PHBV nanofibers containing the highest amount of NPs (15 wt%) showed reduced Young’s modulus and strength, which was probably because of the agglomeration of the NPs. The tensile Young’s modulus and tensile strength of the hybrid PHB/nHA scaffold were higher than those of the PHB scaffold [85]. With the integration of gelatine with the electrospun PHB/nHA, both the tensile Young’s modulus and the tensile strength slightly decreased compared to those of the PHB/nHA mat. The addition of HA–NPs to the polymer matrix up to 15 wt% resulted in a significant increase in the compressive Young’s modulus and compressive strength of the scaffolds [86]. When the nHA content of the scaffolds reached 20 wt%, a significant decrease was observed due to HA agglomeration. The mechanical properties of laminated nHA/PHB scaffolds are significantly improved in comparison to traditional nHA/PHB and PHB scaffolds [87].

Blending PHB with poly(l-lactide-co-ε-caprolactone) (PLCL) significantly reduced the brittleness of the electrospun fibers and significantly increased the extension to break [88]. The addition of GO to PHBV significantly enhanced the tensile strength, Young’s modulus and percent elongation of the nanofibrous scaffold in comparison with pure PHBV [11]. Compression modulus of PHBV film increased by 25% with the addition of GO nanosheets [65]. Incorporation of 0.7 wt% graphene nanoplatelets in the PHB matrix with uniform dispersion resulted in the enhancement of tensile stress from 7.5 to 12.2 MPa [89]. Addition of mechanically stable GO into the polymeric matrix bestowed the flexibility of the PHBV based Fe_3_O_4_/GO-g-PHBV composite enhancing tensile strength and elongation at break [90]. The ductility of the PHB nanofiber scaffold significantly improved with addition of 1 wt% SPN increasing the elongation at break by 190% compared to pure PHB scaffold [66].

PHBV/poly lactic-co-glycolic acid (PLGA) and PHBV/PCL membranes have better mechanical properties than pure PHBV membranes. The highest values of tensile strength, elongation at break and Young’s modulus are in 50/50 hybrid membranes [91]. It has been shown that PCL is able to increase the tensile strength and elongation of PHB, although if the mass ratio of PHB/PCL was higher than 40:60, the effect was not considerable [64]. CTS-g-PCL/PHBHHx fibers possess increased tensile strength, elongation at break and Young’s modulus values compared to pure PHBHHx fibers [92].

PHB electrospun nanofibers have improved tensile strength after MWCNT incorporation and hot-stretching treatment [93]. Maximum values of the tensile strength, breaking elongation rate, initial Young’s modulus and fracture energy of the CMWCNT-g-PHB/PHB composite nanofiber scaffolds are achieved at a CMWCNT content of 6 wt% [68]. The tensile strength and breaking elongation rate of composite nanofiber scaffolds were more than twice those of pure PHB nanofiber scaffolds. CNTs can significantly increase the tensile strength and Young’s modulus of scaffolds. The highest strength and Young’s modulus values are obtained for PHB/0.5wt% CNT nanocomposite scaffolds [16]. A significant tensile strength increase in PHBV/MWCNT nanocomposites was observed upon the addition of MWCNTs with the maximum tensile strength at 1 wt% MWCNT content [94]. The tensile strength of the composite PHB/CNT scaffold was significantly increased in the presence of 1% CNTs compared to pure PHB scaffold [67]. Small amount of the humic acid loaded CNT (HACNT) greatly improved the ductility of the HACNT/PLA/PHB composite, with the maximum tensile strength increased by 236% and the elongation at break improved by 790% [95].

The Young’s modulus and elastic modulus values increased after the addition of nBG to the PHB film but decreased after the addition of mBG [35]. The reduction in Young’s modulus is due to poor mixing of mBG particles with the polymer matrix, leading to large agglomerations. The maximum tensile strength and Young’s modulus were obtained for the PHB/nBG scaffolds containing 7.5 wt%. nBGs. A further increase in nBG content worsened the tensile strength of the nanocomposites due to the agglomeration of nBGs in the polymer matrix at 10 wt%. and 15 wt%. Mesoporous bioglass (MBG) increases the compressive strength of the PHBHHx film with increasing MBG content [96]. The compressive strength of PHBV/mBG composite scaffolds was significantly higher than that of pure PHBV scaffolds [97]. It has also been reported that MBG did not obviously influence the compressive strength of PHBHHx scaffolds [98].

The orientation of the electrospun fibers directly affects the mechanical strength of the scaffolds [99]. Overall, aligned electrospun nanofibers reveal better mechanical properties than random nanofibers.

Deposition of NPs into a polymer matrix also enhances the mechanical properties of the composites [24,82,83,84,85]. The homogeneous dispersion of NPs provides a high interfacial surface, which may enhance the load transfer between the polymer matrix and the NPs, which results in improvement of the mechanical properties of the composite scaffolds. When the concentration of the NPs is low, the matrix can transfer the concentrated stress to the NPs effectively, thus improving the strength of the material. However, as the concentration of the NPs increased, the NPs agglomerated in the polymer matrix, which might weaken the stress transference [84,86]. They act as weak points in the structure and can easily break when stress is applied to the composite. Broken agglomerates then act as stress concentrators leading to the formation of microcracks, consequently leading to a significant decrease in the Young’s modulus and strength of the composites [100,101].

The most important mechanical properties of the hybrid PHA-based composites are summarized in Table 1; Table 2.

For the effective use of PHA-based hybrids, their mechanical properties must fit into the range of native tissue values. Hybrids containing CTS, SF, and collagen have tensile strength and Young’s modulus in the range of those of human cancellous bone and cartilage, making them suitable for bone and cartilage tissue engineering in terms of mechanical properties. The suitability of materials for their use in tissue engineering in terms of cellular reactions will be discussed in the future.

### 2.3. Biodegradation of the Hybrids

The degradation of a biomaterial that is not harmful to the body is known as biodegradation. The products that are produced must be nontoxic to body fluids and cause noninflammatory effects. The advantages of biodegradable materials are that their subsequent removal from the patient’s body is not required and therapeutic agents are easier to deliver locally. The degradation rate depends on the composition of the polymer, its crystallinity, molecular weight, thickness, surface properties and environmental conditions [105]. The biodegradation rate of PHAs is very slow, which makes them good candidates for long-term tissue engineering applications. Since PHAs are biodegradable, their composites with other materials must also be biodegradable and have the necessary mechanical strength to support regeneration of newly formed tissue for a long time [106,107]. Incorporation of different materials into the polymer and fabrication of hybrid PHA-based biomaterials help to achieve a suitable biodegradation rate for different biomedical applications.

The addition of CTS as the hydrophilic counterpart favors the biodegradation of hydrophobic PHAs. The in vitro degradation tests under physiological conditions revealed a very low degradation rate of the PHBV fibrillar mats, with approximately 95% mass retention after 28 days and increasing biodegradability of the hybrid mats as the CTS content increased. At higher CTS contents, water penetration proceeds faster and the loss of fiber integrity occurs very quickly [108]. Electrospun PHB/CTS fibrous scaffolds containing 15 wt% and 20 wt% CTS possessed an enhanced mass loss rate compared to pure PHB [32]. SPN have been shown to be able to accelerate the biodegradation rate of PHB nanofiber scaffold with weight loss percentage increasing from 14.4% to 30.4% after degradation for 30 days in enzyme solution [66]. It has been shown that addition of graphene nanoplatelets into PHB matrix led to decrease in the rate of degradation due to the non-degradable nature of graphene [89]. Incorporation of 20% curcumin increased the biodegradation rate of PHB/MWCNT electrospun nanofibers to about 35% of mass loss after 4 weeks compared to 6% mass loss of neat PHB nanofibers [81].

The weight loss of the PHB/nBG scaffolds with different weight ratios of nBG (0, 2.5, 5, 7.5, and 10 wt%) and various porosities (70, 80 and 90% NaCl) increased with increasing volume fraction of porosity and nBG concentration [109]. Electrospun PHB/cellulose acetate (CA) blend nanofiber scaffolds have a higher degradation rate than neat PHB [110]. The addition of HA into the nanofibrous PHBV film led to a very high specific surface area per volume and thus enhanced the degradation rate [57]. PLA/PHB electrospun mats with the addition of acetyl tri-n-butyl citrate (ATBC) plasticizer showed a higher degradation rate compared to neat PHB mats [111].

For PHA-based materials to be used in tissue engineering, the rate of degradation should match the rate of tissue regeneration. Therefore, the use of various fillers in PHA-based composites that affect the rate of degradation makes it possible to expand the range of applications of these materials in tissue engineering.

### 2.4. Piezoelectric Properties

Piezoelectricity was discovered in 1880 in quartz crystals and Rochelle salt led by Curie brothers [112]. Fukada and Yasuda discovered a piezoelectric effect in bone in 1957 [113] and inspired extensive studies on electromechanical effects in bone and their role in modulating cellular behavior to control growth and bone regeneration processes [114,115,116]. Piezoelectric materials can generate electrical charge in response to deformations and vice versa. This effect occurs as a result of the formation of a net dipole moment and subsequent polarization of the material [117]. Piezoelectricity can also occur due to poling or aligning of dipoles within a material in a sufficiently high electric field [118]. Some piezoelectric materials are initially nonpolar and generate charge only under stress. Others are permanently polar, and they have net dipole moments without any force application. The piezoelectric charge constant (d_ij_ constant) is an expression of the amount of charge that the material generates in response to stress applied or alternatively represents the strain experienced by the material per unit electric field applied. These materials can deliver an electrical stimulus to cells to promote tissue formation and regeneration without the need for an external power source [119]. The piezoelectric properties of PHA-based materials are provided by the presence of an asymmetric carbon linked to a polar oxygen group [120]. For pure PHB electrospun nanofibers, its piezoelectric charge constant is approximately 3.25 pC/N for d_31_ and 4.13 pC/N for d_33_. The piezoresponse of polymers can be enhanced with the addition of different nanomaterials as fillers, such as CNTs [93], barium titanate (BaTiO_3_) [121], and PANi [122]. The phenomenon of the increased piezoresponse is due to an increase in the piezoactive phase amount [93]. The β-form crystals have an all-trans conformation in which the dipoles are aligned in the same direction, which is normal to the chain axis. The unit cells of β-phase crystals consist of two all-trans chains packed with their dipoles pointing in the same direction, which allows the largest spontaneous polarization and improved piezoelectric properties. Unfortunately, there are only a few works revealing the piezoelectric properties of hybrid PHA-based biomaterials.

A piezoelectric PHBV/BaTiO_3_ nanohybrid scaffold with 20 wt% BaTiO_3_ showed an enhanced piezoelectric response similar to that of native bone [121]. Electrically polarized scaffolds demonstrated better cellular activity of human mesenchymal stem-cell-derived chondrocytes than unpoled scaffolds. The polarized scaffolds promoted cell attachment, proliferation, and collagen II gene expression in comparison with control (pure PHBV) and unpoled scaffolds. PHB/MWCNT composite nanofiber membrane scaffolds fabricated by electrospinning followed by hot-stretching treatment had enhanced piezoresponse [93]. After blending with MWCNTs and hot-stretching treatment, the piezoelectric charge constant increased to 13.41 pC/N for d_31_ and 15.11 pC/N for d_33_ and 25.71 pC/N for d_31_ and 26.8 pC/N for d_33_, respectively It has been reported that the doping of piezoelectric polymer PHB with conductive PANi allowed an increase in the piezoelectric charge coefficient and surface electric potential of pure PHB scaffolds by 4.2 times and 3.5 times, respectively, for 2 wt% PANi [122].

The piezoelectricity of biomaterials hypothetically may stimulate cell growth and guide axonal growth, making them attractive for tissue engineering applications. Composites with a greater piezoelectric charge constant than neat PHAs have also been reported [93,121,122]. However, the effect of the piezoelectricity of PHA- and PHA-based materials on cell growth is not fully understood and requires additional investigation.

## 3. PHA Based Composites for Tissue Engineering

### 3.1. Bone Tissue Engineering

Biomaterials for bone tissue engineering should fulfil the requirements described in detail elsewhere [9,123]:

Mechanical strength to withstand hydrostatic pressure.Osteoinductivity to promote the migration of osteogenic cells and stimulate differentiation. An important role in osteoinductivity is played by the chemical composition of the scaffold, its porosity, surface properties and nano/microtopography.Porosity to provide delivery of nutrients to cells, remove cellular waste and promote vascularization. Fabrication of porous biocompatible PHA-based materials makes them more suitable for cell growth and allows cells to penetrate into the scaffold. Pore size should be at least 100 μm in diameter for successful diffusion of essential nutrients and oxygen supply. However, pore sizes in the range of 200 to 350 μm were found to be optimal for bone tissue in-growth [9,123].Vascularization to avoid ischaemia and cell apoptosis.Bioresorbability to allow new bone tissue formation. The scaffolds should degrade at a controlled resorption rate, creating space for new bone tissue formation. Degradation products should not cause inflammation to the surrounding tissues.

Figure 2 shows the temporal functions of the bone tissue scaffold after implantation. Biocompatibility, porosity and adequate mechanical strength are essential properties for cell growth, differentiation, mineralization, vascularization and ECM formation processes [28]. The crucial function of the scaffold is its capacity to bioresorb while regeneration of new bone takes place. The role of the polymer scaffolds in tissue regeneration as well as techniques commonly employed for bone-to-tendon interface reconstruction is well described in critical review [124].

Recent studies have shown that incorporation of HA into different biomaterials (PHBHHx, PLA, PCL, CNTs, and titanium) enhances mechanical performance and osteoblast response [82,125,126,127,128,129]. PHB/nHA scaffolds with 10 and 15 wt% nHA content were biocompatible with MG-63 cells in the indirect method of cytotoxicity evaluation [86]. In addition, the morphology of the attached MG-63 cells in direct contact with the scaffolds indicated the appropriate cell–scaffold interaction. According to a previous study [130], the addition of 15 wt% HA–NPs to the PHB matrix led to better attachment, spreading, and proliferation and a significant increase in the metabolic activity of 3T3 and MC3T3-E1 cell lines compared to the neat polymer. HA induced the differentiation of MC3T3-E1 cells, which also exhibited an elongated shape with an increase in the expression of cytoskeletal F-actin. HA blended with PHB improved osteoblast cell growth and alkaline phosphatase (ALP) activity compared to neat PHB scaffolds [82]. It has also been reported that the addition of nHA into PHBHHx scaffolds fabricated by salt leaching had no effect on their mechanical properties or osteoblast responses. A hybrid tri-layered scaffold PHB/nHA conjugated with the modified gelatine/nHA hydrogel was prepared [85]. A scheme of tri-layered PHB/nHA scaffold fabrication and subsequent in vitro test visualization are shown in Figure 3. PHB/nHA provided the required mechanical strength, while the hydrogel acted as a cell carrier and provided the ECM-like environment necessary for the cells to spread and proliferate. Bone cells inside the electrospun nanofiber scaffolds were highly viable and infiltrated into the scaffolds after 14 days of encapsulation. Moreover, encapsulated HA–NPs within the hybrid scaffold effectively increased matrix mineralization.

The best cell growth and differentiation of murine marrow osteoblasts were obtained on PHB/HA scaffolds containing 10 wt% and 20 wt% HA [33]. The nHA-sprayed PHB scaffold promoted the differentiation of human mesenchymal stromal cells (hMSCs) towards the osteoblast phenotype [131]. Human bone marrow mesenchymal stromal cells (hBMSCs) exhibited better adherence, proliferation and osteogenic phenotypes on laminated electrospun nHA/PHB-composite scaffolds than on PHB scaffolds [87]. The laminated scaffold exhibited a loose character compared with the dense morphology of the traditional scaffold (Figure 4B). The fibrous nHA/PHB scaffolds had high porosity, pore interconnectivity, and a large pore size. The nHA/PHB composite fibers were rougher, with nHA NPs protruding from the fibers, while pure PHB fibers were smooth (Figure 4C). In vivo tests showed no inflammatory reactions, infections, or extrusions. Blood vessels were formed and clearly observed on the surface of the scaffolds (Figure 4D). The staining in the nHA/PHB group was much denser than that in the PHB group, indicating more ECM formation in the nHA/PHB group than in the PHB group (Figure 4E). The mean blood vessel density in the nHA/PHB group was significantly higher than that in the PHB group (Figure 4F).

PHB/nHA scaffolds obtained by the thermally induced phase separation technique with in situ nHA incorporation were fully cytocompatible and able to sustain MC3T3-E1 mouse pre-osteoblast adhesion and proliferation. Differentiated cells predominantly had osteocyte-like morphology, which was not observed for neat PHB scaffolds. In situ synthesized nHA-loaded samples had the highest ALP production and typical morphology of the terminal differentiation stages of osteoblasts [132]. Electrospinning was also used to obtain PHBV/nHA/SF composites [133]. In vitro biological tests proved that the composite fibrous membranes were biocompatible and supported human osteoblast cell attachment. The cells penetrated into the composite membrane and elongated themselves in the direction of the fibers after 3 days of culture. Nonwoven electrospun PHBHHx/SF films were fabricated by the electrospinning technique [31]. The differentiation study revealed that the electrospun PHBHHx/SF film supports the differentiation of hUC-MSCs into the osteogenic lineage and enhances their proliferation. Electrospun PHBHHx/SF film upregulated the expression of the osteogenic marker genes ALP and osteocalcin (OCN) by 1.6-fold and 2.8-fold, respectively, after 21 days of osteogenic induction. A composite PHBV/HA coating obtained via a matrix-assisted pulsed laser evaporation technique was investigated [134]. Mesenchymal bone progenitor cells successfully adhered and spread onto the PHBV/HA composite coatings without changes in their morphological features. Substrates supported cell growth during the in vitro osteogenic differentiation of MSCs.

The PHBV/CTS/8 wt% nHA scaffold had enhanced osteoblast proliferation and higher ALP and mineral deposition than the PHBV scaffold due to the synergistic effect of CTS and nHA, whereby CTS provided cell recognition sites, while nHA acted as a chelating agent for organizing apatite-like mineralization [135]. The presence and distribution of calcium deposits on the cell-scaffold constructs is presented in Figure 5. The amount of minerals increased by day 20 (Figure 5(A2–E2)) compared to the mineral deposition by day 5 (Figure 5(A1–E1)). Cells on PHBV/CTS/nHA8 scaffolds showed brighter red stains than on the other scaffolds on both day 5 and day 20, indicating that the osteoblasts seeded on such scaffolds secreted more minerals with the cooperative interaction of CTS and nHA.

Figure 6 reveals that the osteoblast cells acquired a more homogeneous distribution on PHBV/CTS/nHA scaffolds and preserved the phenotypic expression of bone-specific protein OCN at higher levels than those inoculated on PHBV and PHBV/CTS scaffolds. The increased level of OCN expression was associated with the incorporation of CTS and HA, promoting cell growth and mineral-rich matrix deposition and supporting osteoconduction.

Collagen is the most abundant protein (by weight) in animals, accounting for 30% of all proteins in mammals, and represents an important protein for anchoring cells, such as fibroblasts or epithelium. Collagen assembles into different supramolecular structures and has exceptional functional diversity. Collagen is the major protein of connective tissue, tendons, ligaments, and the cornea, and it forms the matrix of bones and teeth [136]. Depending on the application, the natural protein can be grafted or dip coated on the polymer surface [137,138]. Collagen has the potential to serve as a biomaterial for bone tissue engineering due to its abundance, biocompatibility, high porosity, facility for combination with other materials, easy processing, hydrophilicity, and low antigenicity [139].

Collagen type I was immobilized on the surface of the porous PHBV/nHA composite scaffold [140]. It has been reported that PHBV/nHA/Col composite scaffolds exhibit significantly higher osteoblast cell differentiation and proliferation and better adhesion as well as better production of ALP than PHBV/nHA composite scaffolds and PHBV scaffolds. The morphologies of osteoblasts on the surface of the three scaffolds after 4 h, 1 day, and 2 days of incubation are shown in Figure 7. Osteoblasts adhered more quickly to the collagen-immobilized PHBV/HA scaffold than to the PHBV/nHA and PHBV scaffolds.

The proliferation and morphology of bone cells (mouse osteoblastic cell line MC3T3-E1 subclone and rat osteosarcoma cell line UMR-106) cultured on physically and chemically immobilized collagen PHBV surfaces in comparison with untreated PHBV films was investigated [17]. The bone cell activity on chemically and physically immobilized collagen PHBV films was found to be 246 and 107% for UMR-106 and 68 and 9% for MC3T3 cell lines, respectively. The chemically immobilized collagen on the PHBV surface provided a more favorable matrix for cell proliferation.

Among all the bioactive materials, the most prominent bioactive behavior belongs to BGs containing a group of compounds that connect tissues in a short time [141]. The good ability of BGs to bind to their surrounding tissues has made them an interesting and popular subject of research. This property of BG is actually due to the formation of a layer of hydroxyapatite carbonate on the surface of glass. BGs have a higher level of biocompatibility than calcium phosphates [142]. In addition, these BGs prevent the formation of root tissue at the implant–bone interface and encourage the formation of a strong chemical bond between the implant and bone tissue. PHBHHx/MBG composite scaffolds obtained by a 3D printing technique exhibited good apatite-forming bioactivity and stimulated HBMSC adhesion, proliferation and differentiation [96]. In vivo experiments revealed that PHBHHx/MBG composite scaffolds had good osteogenic capability and stimulated bone regeneration in critical-size rat calvarial defects within 8 weeks.

PHB/7.5 wt % nBG scaffolds exhibited better MG63 cell line attachment and osteoconductivity and significantly improved cell proliferation compared to pure PHB scaffolds [109]. The addition of 10 wt % nBG to the PHB matrix increased MG-63 osteoblast cell proliferation, while the same amount of mBG decreased proliferation. Further addition of bioactive glass reduced cell proliferation [35]. Mesoporous bioglass-doped PHBHHx (PHBM) composite scaffolds had higher ALP activity levels, cell viability and growth rates of hMSCs than pristine PHBHHx scaffolds. The distribution of the cells on the PHBM composites is shown in Figure 8. ALP staining on the PHBM-8 (8 wt % of MBG) surfaces was significantly denser than that on the other sample surfaces at day 14 [98].

The addition of mBG to PHBV cocultures of human umbilical vein endothelial cells (HUVECs) and HBMSCs enhances osteogenic differentiation of cocultured HBMSCs and vascularization of cocultured HUVECs by upregulating paracrine effects between the two types of cells compared to pure PHBV scaffolds [97]. The effect of different scaffolds on the vascularization of cells is shown in Figure 9 (capillary-like networks are shown with white arrows). Greater amounts of HUVECs were observed in the composite scaffolds than in the pure PHBV scaffolds at 14 and 28 days. Among all groups, composite scaffolds containing PHBV with 10 wt % mBG showed the strongest stimulatory effects on osteogenic differentiation and vascularization. In vivo results also demonstrated that PHBV containing 10 wt % mBG scaffolds with cocultures of HUVECs and HBMSCs showed the strongest stimulatory effects on osteogenesis and angiogenesis among all the groups.

With increasing CMWCNT content in electrospun PHB/CMWCNT-g-PHB composite nanofiber membrane scaffolds, the ALP relative activity increased. Scaffolds showed no cytotoxic effects. MG-63 osteoblasts cultured for 7 days were capable of spreading and proliferating [68]. PHB/CNT composites were obtained with significantly improved bioactivity, viability and proliferation of MG63 cells compared to neat PHB scaffolds [16]. In another work, curcumin loaded PHB/MWCNT electrospun scaffolds forced mesenchymal stem cells to differentiate towards osteoblasts. In vivo biocompatibility studies revealed that curcumin strongly reduced inflammatory reaction after 8 weeks of implantation of the scaffolds [81].

Thus, summarizing the results obtained, it can be concluded that the use of HA, BG, and collagen as fillers in PHA-based composites improves bone cell adhesion, proliferation, growth and osteogenic differentiation, which makes them promising for use in bone tissue engineering. However, the use of hybrids is not limited to bone tissue engineering, and they can also be used in cartilage, nerve and skin tissue engineering.

### 3.2. Cartilage Tissue Engineering

Adult articular cartilage is a highly specialized connective tissue that covers the epiphyseal surface of the articular bones, providing mobilization of joints without friction and providing resistance to compression [143,144]. It is an avascular tissue consisting of mature chondrocytes with a heterogeneous ECM [145]. Adult articular cartilage tissue exhibits a limited inherent capacity for regeneration and repair, and cartilage tissue defects often lead to osteoarthritis, ultimately necessitating total joint replacement [146]. Cartilage tissue engineering is a novel approach that utilizes a combination of cells, scaffolds, and growth factors to regenerate lost or damaged cartilage tissue to overcome restrictions of conventional clinical methods [147].

CTS can support chondrogenic activities and facilitate articular cartilage repair [148,149]. Electrospun PHB/CTS fibrous scaffolds containing 15 wt % and 20 wt % CTS possessed good biocompatibility for cartilage tissue [32]. Chondrocytes attach, spread and penetrate more effectively into the PHB/CTS scaffolds than neat PHB scaffolds. SEM images of seeded chondrocytes on the PHB/CTS scaffolds are shown in Figure 10. In the case of pure PHB scaffolds, the cells attached to the fibers did not spread well. PHB/CTS blend scaffolds were more appropriate than pure PHB scaffolds because chondrocytes were spread and penetrated into the polymer matrix of the fibers.

Porous PHB/CTS scaffolds were fabricated and investigated [150]. The results demonstrated that PHB/CTS scaffolds supported chondrogenic differentiation of rat MSCs in vitro. Acellular PHB/CTS scaffolds were successfully utilized in vivo for the repair of artificially created knee cartilage defects in sheep and supported wound healing and the formation of hyaline cartilage-like tissue. Chondrocyte viability on the PHB/CTS/1 wt % MWCNT scaffold was higher than that on neat PHB. Composite scaffolds had proper cell attachment and adequate cellular proliferation and distribution on the scaffold’s surface [54].

Rabbit articular chondrocytes were seeded into PHBV and PHBV/nBG scaffolds, and cartilage regeneration in vivo was evaluated [151]. The incorporation of BG into PHBV efficiently improved both the hydrophilicity of the composites and the percentage of adhered chondrocyte cells and promoted cell migration into the inner part of the constructs.

Thus, based on the above literature survey, it can be concluded that CTS and BG incorporation into PHA matrices improves cellular adhesion, proliferation and chondrogenic differentiation of such materials. As-seeded chondrocytes and growth factors may be employed for better regeneration of cartilage tissue defects.

### 3.3. Nerve Tissue Engineering

Nerve regeneration following nerve tissue injury remains a major issue in the therapeutic medical field. Various biomimetic strategies are employed to direct nerve growth in vitro, among which the chemical and topographical cues elicited by the scaffolds are crucial parameters that are primarily responsible for axon growth and neurite extension involved in nerve regeneration. During the regeneration process, axons are guided to their targets by the cues provided by the substrate, which could be either topographical or chemical and biological cues or their combinations [48]. Therefore, an ideal scaffold to promote the functional recovery of nerves after injury should support axonal growth and guidance. Electrospun nanofibers can mimic the temporal cellular environment and provide signaling cues to the cells in direct contact. Such interactions could guide cellular activities such as proliferation, migration and differentiation.

A 40.01% and 5.48% higher proliferation of nerve cells (PC12) on aligned PHBV/Col50:50 nanofibers compared to cell proliferation on aligned PHBV and PHBV/Col75:25 nanofibers was observed, respectively [14]. Aligned nanofibers of PHBV/Col provided contact guidance to direct the orientation of nerve cells along the direction of the fibers, thus endowing elongated cell morphology, with bipolar neurite extensions required for nerve regeneration. SEM images of electrospun random and aligned PHBV/collagen nanofibers are shown in Figure 11. Optimizing the electrospinning conditions, either aligned or random nanofibers can be obtained.

Figure 12 shows the interaction of PC12 cells with the electrospun random and aligned nanofibers observed by SEM. Very few cells were observed on PHBV nanofibers, and the cells on the PHBV nanofibers had lost their morphology (Figure 12A,E). All the available surfaces of PHBV/Col scaffolds were covered with cells (Figure 12B,C,F,G). The cells on aligned nanofibers were found to orient along the direction of orientation of the fibers, and the cells had a much more elongated morphology compared to the cells on random fibers (Figure 12F,G).

Figure 13 shows the interaction of PC12 cells with the electrospun random and aligned nanofibers observed by immunocytochemical analysis. The cells on PHBV appeared rounded (Figure 13A,E), while the cells on random PHBV/Col nanofibers expressed NF200 (Figure 13B,C) with a phenotype comparable to that of cells on TCP (Figure 13D). The cells on the aligned PHBV/Col nanofibers followed the directional path provided by the nanofibers, and the cells oriented the neurite to touch and interact with the nearby cells (Figure 13F,G).

Cellular investigations with Schwann cells showed nontoxic behavior and better adhesion, growth and viability on collagen cross-linked nanofibrous mats than on other samples [13]. The cells were spread and adhered better on the surface of collagen cross-linked PHB scaffolds.

It has been reported that PCL–PHB/heparin scaffolds with a heparin concentration of 20 mg/mL favored the regulation of induced pluripotent stem cells to differentiate towards neurons and reduced the residual phenotypic iPS cells [152].

Thus, it can be concluded that neat PHAs have poor adherence to nerve cells. Collagen-modified PHA-based materials have improved the cellular response of nerve cells. However, not only the composition of the materials but also the morphology of the composites affects the cellular activity. Fabrication of aligned nanofibrous scaffolds is a simple way to improve adherence and spread of nerve cells.

### 3.4. Skin Tissue Regeneration and Wound Healing

Wound healing is a very complex and interactive process that combines multiple cell types, including dermal and epidermal cells, immune cells, ECM, plasma-derived proteins and growth factors, in the regeneration of skin tissue [153,154]. Scaffolds prepared from both bioactive natural and synthetic polymers are good candidates for wound dressing materials due to their unique properties, including ability to support cell growth, excellent biocompatibility, strength/durability and controlled degradation [155]. A common approach has been the use of biodegradable scaffolds to sustain and guide cell growth through the regeneration process. The role of the scaffold is to provide an artificial environment enabling cell adhesion, migration, and spreading, thus allowing cell proliferation and ECM synthesis. The essential cell type involved in the wound healing process is fibroblasts. They play a critical role in the proliferative phase because they are involved in the production of ECM constituents and growth factors, which are crucial in the subsequent phases of wound healing [156].

Excellent fibroblast proliferation in PHB/CTS scaffolds was observed [157]. PHBV/CTS electrospun nanofibrous mats to develop scaffolds for skin regeneration were fabricated, and L929 cells were used to evaluate fibroblast adhesion, vitality and proliferation. PHBV/CTS 4:1 (*w*/*w*) exhibited a higher in vitro biocompatibility and a better ability for fibroblast adhesion and growth than PHBV/CTS 2:3 (*w*/*w*) [108]. In vivo studies showed good performance of the scaffolds in the wound healing process in rats. Organic soluble PHB/CTS ultrafine fiber membranes were fabricated by electrospinning, and cytotoxicity, mouse fibroblast (L929 cell line) attachment and proliferation were studied. PHB/CTS ultrafine fibers promoted good cell attachment, growth and proliferation and exhibited nontoxic behavior [158]. Electrospun PHB/PANi composite scaffold modified by air plasma and printed by an inkjet method enhanced the proliferation and migration of mouse fibroblast (L-929) and human dermal fibroblast cell lines [77]. SPN-modified PHB nanofiber scaffold presented better cytocompatibility, provided cell-substrate interactions favorable for better cell adhesion and proliferation of NIH3T3 mouse fibroblast cells and MG-63 human osteoblast cells [66].

Chinese hamster lung fibroblast cell adhesion studies on PHB/bacterial cellulose (BC) nanocomposites revealed that cells incubated with nanocomposite scaffolds for 48 h were capable of forming cell adhesion and proliferation, which shows much better biocompatibility than pure PHB [159]. Scaffolds with 2 wt % BC possessed higher biocompatibility with mouse L929 cells than pure PHB [15]. Electrospun PHB/CA blend nanofiber scaffolds have much better biocompatibility than pure PHB films for 3T3 fibroblasts [110]. Cells incubated with the PHB/CA blend scaffold for 48 h were capable of forming cell adhesion and proliferation.

The coaxially electrospun fibers of gelatine-coated PHB exhibited competent tensile properties for skin regeneration with high surface area and porosity. The inclusion of gelatine improved the elasticity of fibers. The fibers supported the growth of human dermal fibroblasts and keratinocytes with normal morphology [160].

Human fibroblasts grow and proliferate better on SF-modified porous PHBHHx scaffolds than unmodified scaffolds [12]. Incorporation of collagen peptides into P(3HB-co-4HB) nanofibers leads to an increase in the proliferation of mouse fibroblast cells (L929). An in vivo study showed that the nanofibrous P(3HB-co-4HB)/collagen peptide construct had a significant effect on wound contractions, with the highest percentage of wound closure of 79%. Photographs of the rat wound treatment are shown in Figure 14. No visible difference in wound appearance was observed for any group on the first day. At day 7 post grafting, tissue formation was clearly observed in nanofibrous P(3HB-co-4HB)/collagen peptide-treated rats. On day 14, wounds treated with nanofibrous P(3HB-co-4HB)/collagen peptides exhibited complete wound closure without the formation of inflamed scabs [161].

Collagen addition into electrospun nanofibrous PHBV/collagen/GO scaffolds increased human dermal fibroblast cell (3T3-L) attachment and reduced cytotoxicity by increasing the hydrophilicity of the scaffolds [11]. Incorporation of GO nanosheets into the PHBV matrix has proven to be a non-cytotoxic approach to enhance adhesion, distribution and proliferation of canine adipose-derived mesenchymal stem cells and provide antibacterial activity against the Staphylococcus aureus [65]. Curcumin and *Gymnema sylvestre* loaded GO/PHB/SA composite increased the cell viability of wounded and diabetic wounded cells without producing cytotoxic effects and enhanced wound closure [162]. Functionalized CNTs have been shown to be able to improve bioactivity, in vitro viability of periodontal ligament stem cells and in vivo tissue compatibility of the PHB electrospun scaffolds [67].

Thus, based on a literature survey, CTS, BC, SF, gelatine and collagen are the most commonly used fillers to improve the performance of PHA-based scaffolds in the wound healing process. The presence of these materials in PHA-based composites accelerates the skin regeneration process and improves cell adhesion, vitality and proliferation.

## 4. Future Prospects and Challenges

The structure-property relationship, composition and mechanical properties (strength, Young’s modulus) of a material are vital factors of its successful biomedical application. Hybrid materials based on PHAs, depending on the fabrication method, reveal a tunable structure, composition, and mechanical properties. However, additional short- and long-term implantation studies of the PHA-based composites described in this study in different animal models in vivo are required to examine their performance to forecast their clinical behavior, which is associated with humans’ daily routine.

The hydrophobic nature of PHAs restricts their widespread biomedical applications due to insufficient cell adhesion or infiltration into the material’s interior when associated with porous scaffolds. To improve the attachment of cells to the surface of a biomaterial and promote cell proliferation, hydrophilicity is an essential property. The surface of PHA-based materials can be turned hydrophilic by fabrication of hybrids using different materials as fillers, such as BG [48], SF [31,50], CTS [32,52], CNTs [68,69] and collagen [11,13], as well as scaffold surface treatments, such as grafting techniques [17,68] and plasma treatment [74,76]. However, some of these methods deteriorate the mechanical properties of the prepared hybrids. Therefore, it is necessary to use approaches for surface and bulk property modification that will improve the hydrophilicity and without impairing the mechanical properties.

Biodegradability is an advantageous factor for tissue regeneration. The degradation rate should match that of new tissue formation. Pristine PHAs have a lower degradation rate than new tissue formation [163,164,165]. The slow degradation rate of PHAs can be altered by blending PHAs with more rapidly degrading materials, such as PLA, CA, and CTS. One of the prospective methods to control the biodegradation rate of PHA without worsening the mechanical properties is UV treatment of PHA powder [166]. However, there is a lack of investigations on the biodegradation of PHA-based materials in terms of their biodegradation and replacement with native tissue in animal models. Further investigations, including in vivo replacement in animals, should be carried out to examine the degradation behavior of the scaffolds to evaluate the possibility of clinical application of these materials in tissue engineering.

The piezoelectric properties of PHA-based hybrid materials are attractive for biomedical applications; however, the impact of piezoeffects on cell behavior or tissue regeneration is still a subject for further studies. There are only a few works aimed at investigating the influence of CNTs [93], BaTiO_3_ [121] and PANi [122] on the piezoelectric response of such materials. In addition, most of the papers are not focused on the biological response of cells but on the physical aspects of the piezoelectric nature of PHA-based composites. Thus, additional biological tests should be performed to investigate the cell response to the PHA-based scaffold piezocharge, whose formation requires additional usage of bioreactors where cyclic loading or deformation can be applied to a piezomaterial. The effect of piezoresponses of different hybrid PHA-based materials, such as on bone tissue regeneration, should also be studied in more detail in vivo under cyclic loading on a PHA-based piezomaterial, and after that, if the effect is of critical importance and is estimated to be proven, further clinical trials should also be performed.

Techno-economic challenges of the developed composites are defined by the limiting factors affecting upscaling of the laboratory samples to make them commercially available. The most important factor is that the criteria to have a choice of the hybrids, the most prospective for a specific biomedical application, should still be defined, since there are challenges to be met in respect with the performance of the composites in the specified application area. As an example, biodegradation rate is a crucial parameter, which defines application of PHAs and their service life-time as implants. Biological properties and biodegradation are strongly affected by the type of the filler used and its content in the PHA matrix [122,167]. In further, the content or concentration of specific filler should be defined to allow wide-spread application of the developed composites. These challenges are still to be overcome in the near future.

PHAs are gaining increasing attention in the biodegradable polymer market due to their promising properties such as high biodegradability in different environments, not just in composting plants, and processing versatility. Among biopolymers, these biogenic polyesters represent a potential sustainable replacement for fossil fuel-based thermoplastics. The most of commercially available PHAs are obtained with pure microbial cultures grown on renewable feedstocks (i.e., glucose) under sterile conditions but recent research studies are focused on the use of wastes as growth media. PHA can be extracted from the bacteria cell and then formulated and processed by extrusion for production of rigid and flexible plastic [168] and also used for biomedical applications [169].

## 5. Conclusions

In this study, hybrid biomaterials based on PHAs with different inorganic fillers (HA, BG, and BaTiO_3_), PHA blends with other polymers (PLA, PCL, PLCL, PLGA, CTS, and PANi) or bioactive components (SF, gelatine, and collagen) and variations in their properties, depending on the type of filler material, were discussed. In vitro tests have shown that the PHA-based hybrid materials presented above are biocompatible, provide adhesion, proliferation and differentiation of various cells, including osteoblasts, fibroblasts and chondrocytes, and reveal enhanced mechanical properties to bear loading and withstand stress in vivo. The future challenges and potential use of PHA-based composites for biomedical applications, such as bone, cartilage, nerve and skin tissue engineering, are revealed. Elaboration of the routes to control surface wetting behavior, in particular, those providing hydrophilic properties (e.g., oxygen or nitrogen plasma treatment procedures) are discussed. This review also revealed the necessity of additional investigations of cell and tissue responses to the piezoelectric charge generated in PHAs-based materials due to their intrinsic piezoelectric properties.

## Figures and Tables

**Figure 1 polymers-13-01738-f001:**
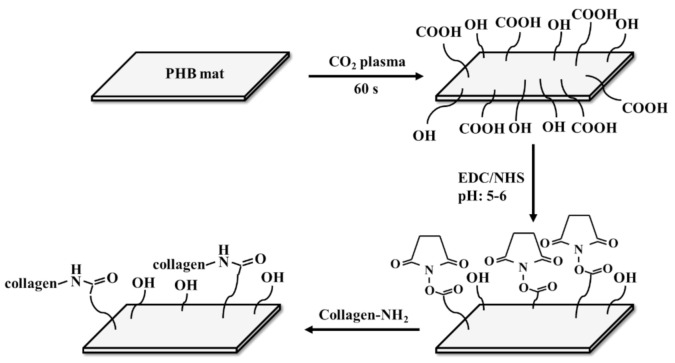
Schematic of the covalent coupling reaction for the attachment of protein to PHB fibers. Re-designed based on [13].

**Figure 2 polymers-13-01738-f002:**
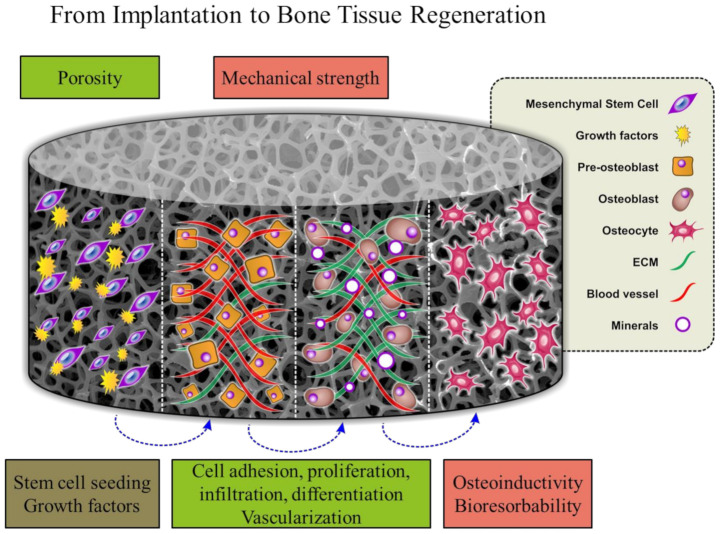
Bone formation followed by scaffold implantation. Reproduced from [28] with permission from Wiley.

**Figure 3 polymers-13-01738-f003:**
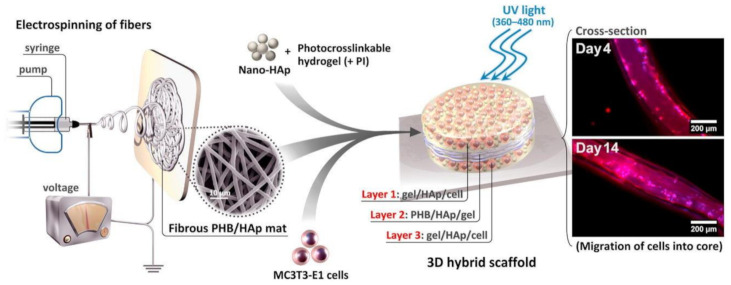
Scheme of tri-layered PHB/nHA scaffold fabrication and in vitro test visualization. Reprinted from [85] with permission from Elsevier.

**Figure 4 polymers-13-01738-f004:**
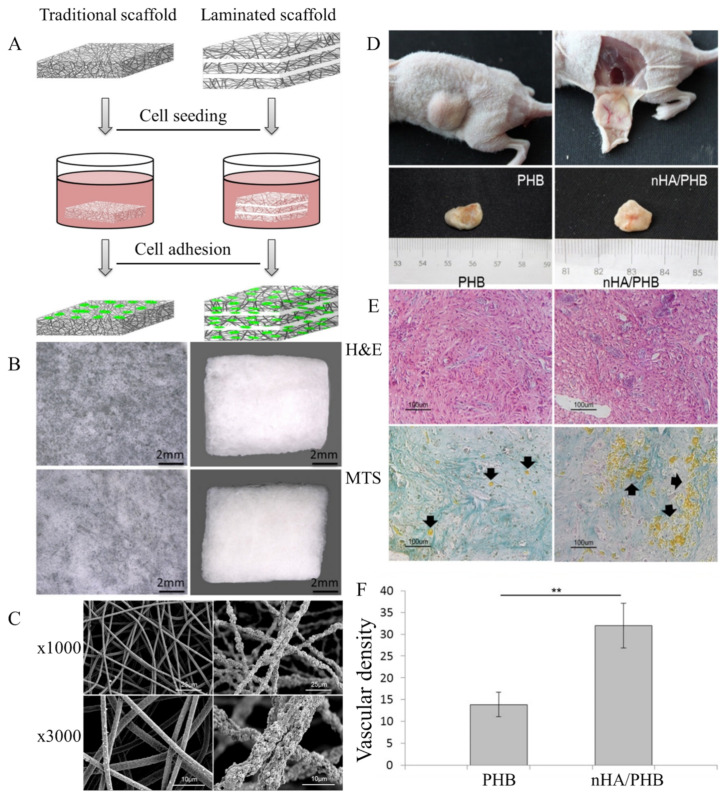
(**A**) Schematic diagram of the traditional electrospun scaffold and laminated electrospun scaffold. (**B**) Surface features of a traditional electrospun scaffold, a thin-layer fiber membrane and a laminated scaffold. (**C**) SEM micrographs of the structure of the laminated electrospun PHB scaffold and the nHA/PHB scaffold. (**D**) Representative macroscopic images of PHB-hMSCs and nHA/PHB-hMSCs composites removed from nude mice after 2 months. (**E**) Appearance of specimens 2 months post-transplantation for the PHB-hMSC composites and nHA/PHB-hMSC composites. Histologic staining with hematoxylin and eosin (H&E) and Masson’s trichrome staining (MTS). Black arrows indicate vascular structures. (**F**) Number of vessels in the specimens 2 months post-transplantation, ** *p* < 0.01. Reprinted from [87] with permission from Elsevier.

**Figure 5 polymers-13-01738-f005:**
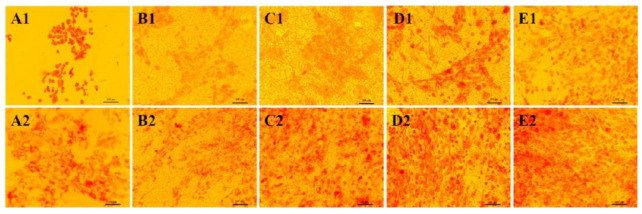
Alizarin Red-Staining (ARS) staining of osteoblasts on Tissue Culture Plates (TCPs) (**A1**,**A2**) and PHBV (**B1**,**B2**), PHBV/CTS (**C1**,**C2**), PHBV/CTS/nHA4 (**D1**,**D2**), PHBV/CTS/HA8 (**E1**,**E2**) scaffolds on days 5 (**A1**–**E1**) and 20 (**A2**–**E2**). Reprinted from [135] with permission from Elsevier.

**Figure 6 polymers-13-01738-f006:**
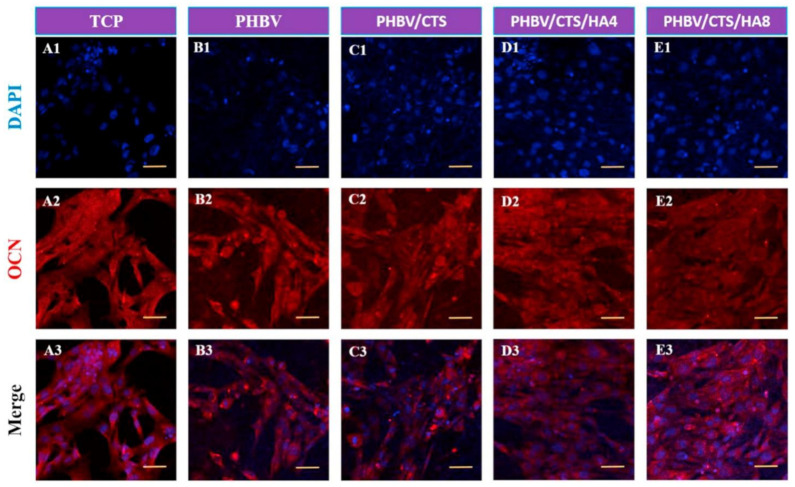
**DAPI** staining (**A1**–**E1**), expression of OCN (**A2**–**E2**) by human fetal osteoblasts and merge images (**A3**–**E3**) after culturing of cells for 5 days on TCP (**A1**–**A3**), PHBV (B1-B3), PHBV/CTS (**C1**–**C3**), PHBV/CTS/4 wt% nHA (**D1**–**D3**) and PHBV/CTS/8 wt% nHA (**E1**–**E3**). Reprinted from [135] with permission from Elsevier.

**Figure 7 polymers-13-01738-f007:**
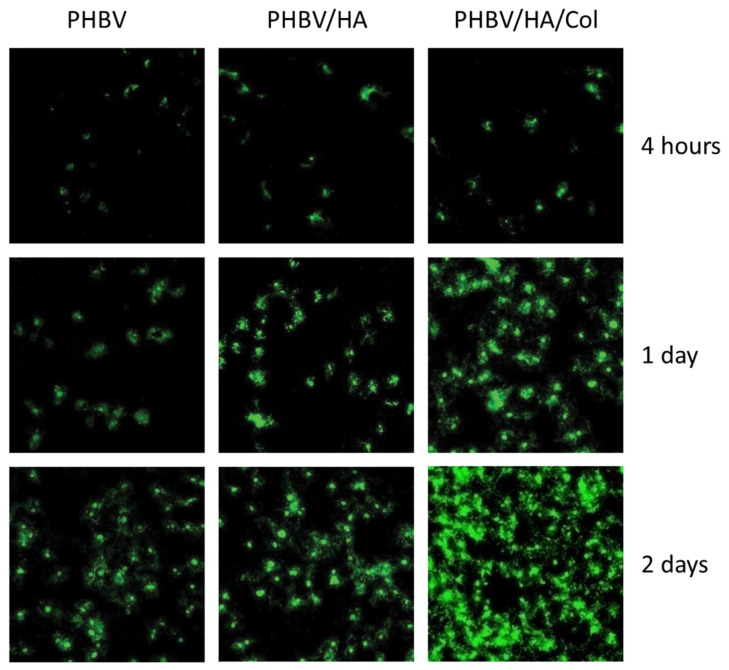
Confocal laser scanning microscopy images of calcein-AM dye-stained osteoblasts on PHBV, PHBV/nHA and PHBV/nHA/Col scaffolds for 4 h, 1 day, and 2 days. Reprinted from [140] with permission from Hindawi Publishing Corporation. This is an open access article distributed under the Creative Commons Attribution License, which permits unrestricted use, distribution, and reproduction in any medium.

**Figure 8 polymers-13-01738-f008:**
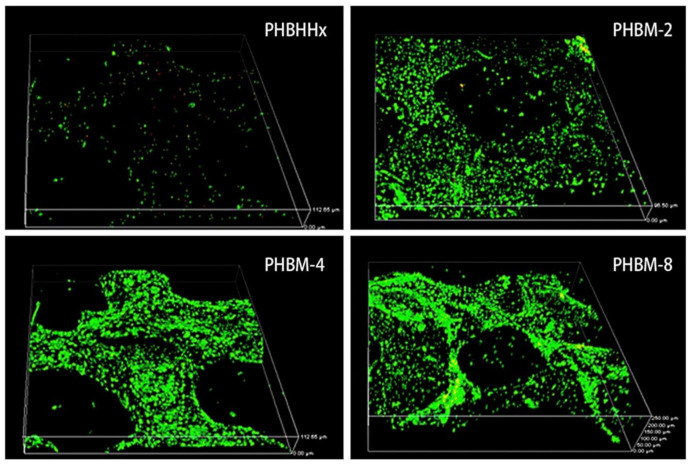
Live/dead staining of hMSCs cultured for 14 days. Live hMSCs were stained green, and dead cells were stained red. Reprinted from [98] with permission from Elsevier.

**Figure 9 polymers-13-01738-f009:**
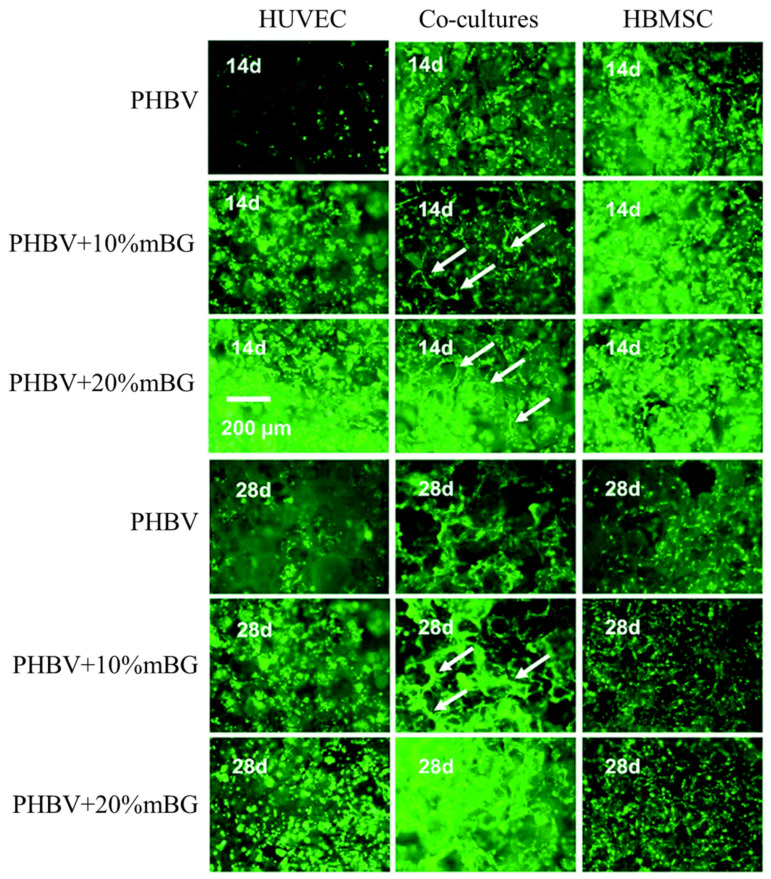
Live-dead staining of cells cultured in different scaffolds for 14 and 28 days. Reproduced from [97] with permission from the Royal Society of Chemistry. This article is licensed under a Creative Commons Attribution 3.0 Unported Licence.

**Figure 10 polymers-13-01738-f010:**
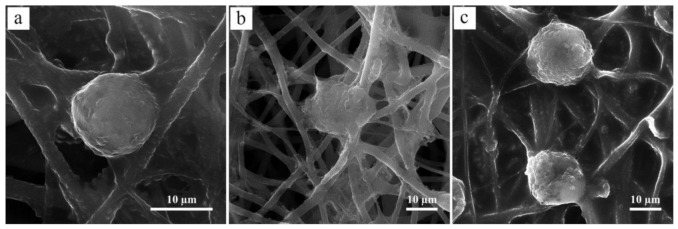
SEM images of cultured chondrocyte cells on the surface of (**a**) PHB, (**b**) PHB/15wt % CTS and (**c**) PHB/20wt % CTS scaffolds. Reproduced from [32] with permission from Wiley.

**Figure 11 polymers-13-01738-f011:**
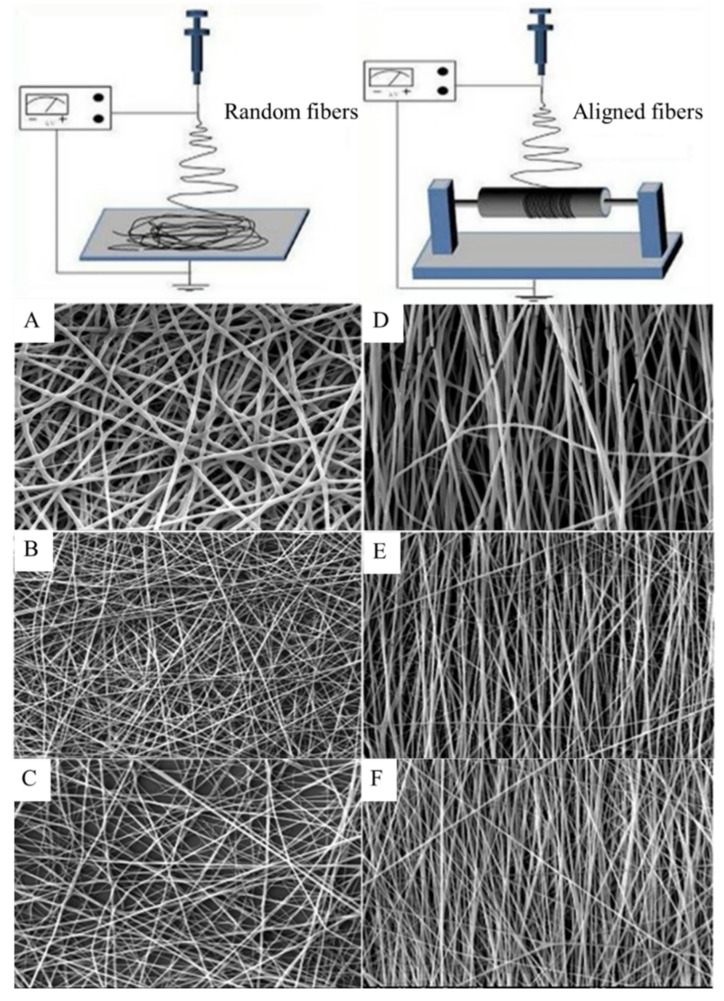
SEM images of electrospun random (**A**) PHBV, (**B**) PHBV/Col75:25, (**C**) PHBV/Col50:50 and aligned (**D**) PHBV, (**E**) PHBV/Col75:25, and (**F**) PHBV/Col50:50 nanofibers. Reproduced from [14] with permission from Wiley.

**Figure 12 polymers-13-01738-f012:**
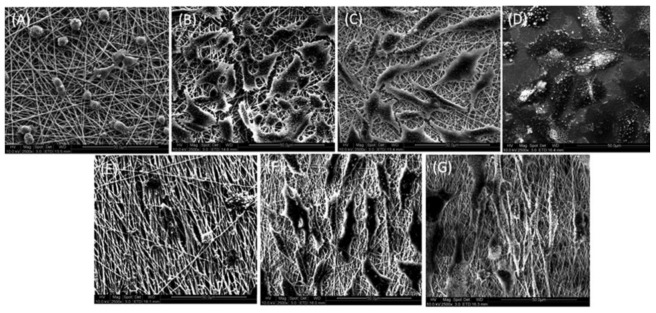
SEM images representing the interaction and orientation of cells on random (**A**) PHBV, (**B**) PHBV/Col75:25, (**C**) PHBV/Col50:50, (**D**) TCP, and aligned (**E**) PHBV, (**F**) PHBV/Col75:25, and (**G**) PHBV/Col50:50 nanofibers. Reproduced from [14] with permission from Wiley.

**Figure 13 polymers-13-01738-f013:**
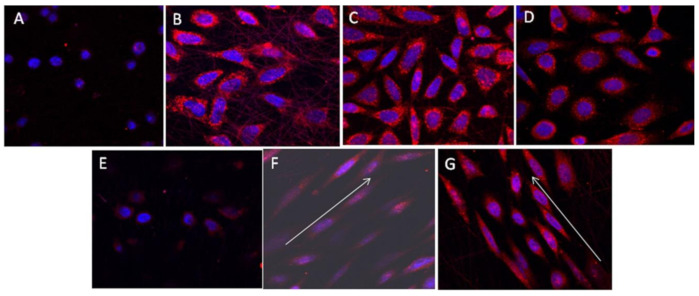
Expression of NF200 by cells seeded on random (**A**) PHBV, (**B**) PHBV/Col75:25, (**C**) PHBV/Col50:50; (**D**) TCP; and aligned nanofibers of (**E**) PHBV, (**F**) PHBV/Col75:25, and (**G**) PHBV/Col50:50. Reproduced from [14] with permission from Wiley.

**Figure 14 polymers-13-01738-f014:**
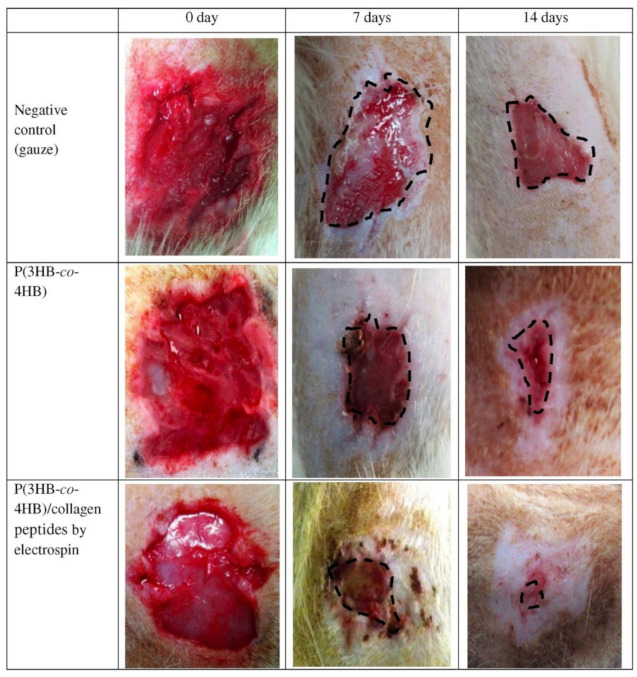
Photographs of wounded skin treated with various P(3HB-co-4HB) at different times. Reprinted from [161] with permission from Elsevier.

**Table 1 polymers-13-01738-t001:** Mechanical properties of the PHA-based composites.

Composite	Tensile Strength (MPa)	Young’s Modulus (MPa)	Elongation at Break (%)	Ref.
PHBV	5.82 ± 0.50	67.7 ± 5.2	50.2 ± 4.5	[51]
50PHBV/50SF	3.87 ± 0.37	60.5 ± 5.0	29.8 ± 2.7	
PHBHHx	11.7 ± 0.5		204 ± 5	[12]
PHBHHx/SF	11.5 ± 0.5		175 ± 5	
PHB	6.23 ± 0.3		11.74	[50]
PHB/SF	3.81 ± 0.1		17.10	
PHB	87 ± 3.02	74.45 ± 2.88	26 ± 1.67	[32]
PHB/10 wt% CTS	63.66 ± 6.10	52.79 ± 4.52	46 ± 4.02	
PHB/20 wt% CTS	31.6 ± 3.37	50.74 ± 2.23	65.5 ± 2.25	
Aligned PHB	16.2 ± 3.11	202.1 ± 97.6	7.3 ± 0.8	[52]
Aligned PHB/15 wt% CTS	8.73 ± 3.65	210.2 ± 90.9	1.45 ± 0.67	
Random PHB	7.6 ± 0.8	164.3 ± 82.4	3.83 ± 0.69	
Random PHB/15 wt% CTS	6.41 ± 3.32	150.8 ± 93.6	1.19 ± 0.71	
PHBV	4.01 ± 0.27	108 ± 2.61	56.34 ± 2.66	[14]
PHBV/Col 50:50	2.17 ± 0.27	70.55 ± 1.78	8.17 ± 1.60	
PHBV	94			[11]
PHBV/GO	254			
PHBV/GO/Collagen	241			
PHB	8.4 ± 1.9	554 ± 25	3.8 ± 1.2	[80]
PHB/CTS	8.7 ± 1.2	467 ± 22	84.1 ± 4.7	
PHB/CTS/BCP	16.5 ± 0.9	524 ± 20	99.2 ± 5.1	
PHB	3.8		11.71	[54]
PHB/CTS	3.4			
PHB/CTS/1 wt% MWCNT	10		20.99	
PHB	10.67 ± 1.01	238 ± 52	7.27 ± 0.49	[58]
PHB/nHA (blend)	16.16 ± 0.86	397 ± 107	12.48 ± 1.57	
PHB/nHA (spray)	5.47 ± 0.18	138 ± 19	4.90 ± 0.25	
PHBV	4.41 ± 0.27	106.70 ± 31.33		[84]
PHBV/10 nHABR	6.35 ± 0.38	158.60 ± 34.67		
PHB	1.2 ± 0.2		10.6 ± 1.4	[88]
PHB/25 wt% PLCL	1.2 ± 0.2		41.6 ± 0.8	
PHBV (100 wt%, *w*/*w*)	0.1	0.34	108.32	[91]
PHBV/PLGA (50:50 wt%, *w*/*w*)	4.65	47	125.65	
PHBV/PCL (50:50 wt%, *w*/*w*)	2.56	20.63	115	
PHBV/PCL (50:50 wt%, *w*/*w*)+ 1 wt% CA	1.55	7.47	210	
PHBV/PCL (50:50 wt%, *w*/*w*) + 10 wt% CA	1.2	7.44	43	
PHB	18.8		7	[64]
40PHB/60PCL	26.9		1358	
PHBHHx	10	220	102	[92]
50PHBHHx/50CS-g-PCL	19	390	148	
PHB	2	108		[16]
PHB/0.5 wt% CNT	5.15	285		
60PLA/40PHB	11.8		43.8	[95]
60PLA/40PHB/0.1wt% HACNT	27.87		346.68	
PHB	12.4			[93]
PHB/MWCNT	16.2			
PHB/MWCNT/hot stretching	21.7			
PHB	1.13 ± 0.021	99.41 ± 2.88		[102]
PHB/7.5 wt% nBG	1.91 ± 1.00	30.59		
Cancellous bone	2–12	20–500		[28,103]
Cortical bone	100–230	3000–30,000		
Cartilages	3.7–10.5	0.7–15.3		[103]

**Table 2 polymers-13-01738-t002:** Compressive mechanical properties of the PHA-based composites.

Composite	Compressive Strength (MPa)	Compressive Young’s Modulus (MPa)	Ref
PHB	22 ± 2	317 ± 70	[82]
PHB/mHA	30 ± 6	419 ± 80	
PHBHHx	8 ± 1	173 ± 49	
PHBHHx/mHA	8 ± 1	68 ± 5	
PHB	2.14 ± 0.11	22.16 ± 2.75	[83]
PHB/10wt% nHA	3.18 ± 0.24	41.33 ± 3.21	
PHB	2.03 ± 0.14	29.06 ± 2.74	[86]
PHB/5 wt% nHA	2.38 ± 0.13	36.91 ± 3.12	
PHB/10 wt% nHA	2.76 ± 0.18	45.73 ± 3.87	
PHB/15 wt% nHA	3.19 ± 0.21	56.12 ± 4.28	
PHBV	0.15 ± 0.02		[97]
PHBV/10 wt% mBG	0.25 ± 0.04		
PHBV/20 wt% mBG	0.32 ± 0.03		
Cancellous bone	2–12	50–500	[104]
Cortical bone	100–200	7000–30,000	

## Data Availability

The data presented in this study are available on request from the corresponding author.

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
