# Peer review of "Review of Hybrid Materials Based on Polyhydroxyalkanoates for Tissue Engineering Applications"

_polymers, 2021, doi:10.3390/polym13111738_

Round 1

Reviewer 1 Report

The review article entitled ‘Review of hybrid materials based on polyhydroxyalkanoates for biomedical applications’. The concept of the review article is interesting and suitable to publish in Polymers Journal. However, in the present form it cannot be accepted and it required substantial major revision.

The major comments are as follows:

1)      Title should be modified in a precise way. Instead of mentioning only biomedical give details

2)      Abstract looks very general and not informative should be rewritten. In abstract authors should mention the importance of research work briefly.

3)      Provide a nice graphical abstract representing the overview of the MS with key highlights. For review articles it should be mandatory and don’t use the figures used in the manuscript.

4)      Introduction looks very general. In the introduction section, write the novelty of the work and the problem statement clearly.

5)      The introduction is lacking about the synthesis of PHA using different carbon sources (pure and complicated waste biomass) that you can refer  and cite recent Bioresource technology 282, 75-80, 2019; Polymers 12 (8), 1704, 2020. Precise research objectives and clear justification of the selection of this review topic is lacking thus major discussion is expected during revision.

6)      Give quantitative data of PHA production throughout the world.

7)      Which metabolic pathways microorganism employed for the synthesis of PHA (PHB, PHB-HB and PHB-Hv) add one paragraph describing the same.

8)      Some investigators are also reported that PHA nanocomposites could be  a viable option for biomedical application refer and cite the recent review article Bioresource Technology Volume 325, April 2021, 124685. Add one section for the same

9)      Add case studies if any reported using PHA hybrid materials would be impressive.

10)  For the applications section add one figure describing all applications of PHA hybrids.

11)  This manuscript lacked substantial discussion of results with the the recent literature authors should concentrate on this during revision. Very few papers have been referred within the year of 2018-2020 add recent studies rel;ated to review articles.

12)  Techno Economic challenges of the developed composites need to be addressed by adding a new section before conclusions.

13)  What are the limitations to use this methodology for commercial application?

14)  The conclusion of the study needs to add with the specific output obtained from the study, it could be modified with precise outcomes with a take home message.

15)  English and grammar mistakes are present. The author should check the manuscript by native English Speaker to improve the quality of the manuscript.

Author Response

  1. Title should be modified in a precise way. Instead of mentioning only biomedical give details.

Answer: Thank you very much for this valuable comment. We have changed the title as the reviewer requested. Now the title reveals the content of the manuscript in a better way.

  1. Abstract looks very general and not informative should be rewritten. In abstract authors should mention the importance of research work briefly.

Answer: Thank you very much for this valuable comment. We have corrected the abstract to clarify the importance of our research.

  1. Provide a nice graphical abstract representing the overview of the MS with key highlights. For review articles it should be mandatory and don’t use the figures used in the manuscript.

Answer: Thank you very much for this valuable comment. The graphical abstract has been modified and now looks more extensive.

  1. Introduction looks very general. In the introduction section, write the novelty of the work and the problem statement clearly.

Answer: Thank you very much for this valuable comment. We have modified introduction section to make our review aims and the challenge statement sound better.

  1. The introduction is lacking about the synthesis of PHA using different carbon sources (pure and complicated waste biomass) that you can refer and cite recent Bioresource technology 282, 75-80, 2019; Polymers 12 (8), 1704, 2020. Precise research objectives and clear justification of the selection of this review topic is lacking thus major discussion is expected during revision.

Answer: Thank you very much for this valuable comment. We have added information about PHA synthesis using different carbon sources in the introduction section. We have also added a new reference as the reviewer recommended.

  1. Give quantitative data of PHA production throughout the world.

Answer: Thank you very much for this valuable comment. We have done additional search and the results revealed that the polyhydroxyalkanoate (PHA) market size is estimated to be USD 62 million in 2020 and is projected to reach USD 121 million by 2025, at a CAGR of 14.2% between 2020 and 2025. The market is mainly driven by the rising demand for PHA industries such as food and packaging services, agriculture, biomedical, and some others. Factors such as consumer awareness about the toxicity of the petroleum based and sustainable ecofriendly bioplastics will drive the PHA market. There are key markets for PHA, which are Europe, followed by North America and Asia, in terms of value and volume [1].

  1. Which metabolic pathways microorganism employed for the synthesis of PHA (PHB, PHB-HB and PHB-Hv) add one paragraph describing the same.

Answer: Thank you very much for this valuable comment. Unfortunately, the synthesis pathways of PHAs by different types of microorganisms are out of the scope of our review.

  1. Some investigators are also reported that PHA nanocomposites could be a viable option for biomedical application refer and cite the recent review article Bioresource Technology Volume 325, April 2021, 124685. Add one section for the same

Answer: Thank you very much for this valuable comment. The works dealing with biomedical application of hybrid PHA-based materials are already presented in our review. We have provided a new reference as the reviewer recommended in the introduction section.

  1. Add case studies if any reported using PHA hybrid materials would be impressive.

Answer: Thank you very much for this valuable comment. The articles presented in our review are case studies that describe the routes to improve the most important properties of PHAs and their application for tissue engineering, which is fully consistent with the topic of our study.

  1. For the applications section add one figure describing all applications of PHA hybrids.

Answer: Thank you very much for this valuable comment. The graphical abstract fully reflects the areas of application of PHA-based hybrid materials.

  1. This manuscript lacked substantial discussion of results with the the recent literature authors should concentrate on this during revision. Very few papers have been referred within the year of 2018-2020 add recent studies related to review articles.

Answer: Thank you very much for this valuable comment. We have extended our review paper with additional 9 articles, addressing hybrid PHA-based materials, the most recently published.

  1. Techno Economic challenges of the developed composites need to be addressed by adding a new section before conclusions.

Answer: We would like to thank the reviewer for this valuable comment. We would also like to emphasize that the focus of this manuscript is on the physical and mechanical and biological performance of the composites, thus we have added the text below to the Challenges section.

“Techno economic challenges of the developed composites are defined by the limiting factors affecting upscaling of the laboratory samples to make them commercially available. The most important factor is that the criteria to have a choice of the hybrids, the most prospective for a specific biomedical application, should still be defined, since there are challenges to be met in respect with the performance of the composites in the specified application area. As an example, biodegradation rate is a crucial parameter, which defines application of PHAs and their service life-time as implants. Biological properties and biodegradation are strongly affected by the type of the filler used and its content in the PHA matrix [2, 3]. In further, the content or concentration of specific filler should be defined to allow wide-spread application of the developed composites. These challenges are still to be overcome in the near future.

  1. What are the limitations to use this methodology for commercial application?

Answer: PHAs are gaining increasing attention in the biodegradable polymer market due to their promising properties such as high biodegradability in different environments, not just in composting plants, and processing versatility. Among biopolymers, these biogenic polyesters represent a potential sustainable replacement for fossil fuel-based thermoplastics. The most of commercially available PHAs are obtained with pure microbial cultures grown on renewable feedstocks (i.e. glucose) under sterile conditions but recent research studies are focused on the use of wastes as growth media. PHA can be extracted from the bacteria cell and then formulated and processed by extrusion for production of rigid and flexible plastic [4] and also used for biomedical applications [5].

The text was added to the manuscript.

  1. The conclusion of the study needs to add with the specific output obtained from the study, it could be modified with precise outcomes with a take home message.

Answer: Thank you very much for this valuable comment. We have modified the conclusion section as the reviewer requested.

  1. English and grammar mistakes are present. The author should check the manuscript by native English Speaker to improve the quality of the manuscript.

Answer: Thank you very much for this valuable comment. We have used Elsevier Language Service to improve the quality of English. The certificate is attached. We also know that MDPI provides language editing before publishing, thus some changes are still possible in the case our manuscript is accepted.

References:

  1. Market, M. A. (2019). Polyhydroxyalkanoate (PHA) Market by Type (Short Chain Length, Medium Chain Length), Production Method (Sugar Fermentation, Vegetable Oil Fermentation, Methane Fermentation), Application, and Region-Global Forecast to 2024. Markets and Markets Research Private Ltd.
  2. Chernozem, R. V., Guselnikova, O., Surmeneva, M. A., Postnikov, P. S., Abalymov, A. A., Parakhonskiy, B. V., Surmenev, R. A. (2020). Diazonium chemistry surface treatment of piezoelectric polyhydroxybutyrate scaffolds for enhanced osteoblastic cell growth. Applied Materials Today, 20, 100758.
  3. Chernozem, R. V., Surmeneva, M. A., Surmenev, R. A. (2018). Hybrid biodegradable scaffolds of piezoelectric polyhydroxybutyrate and conductive polyaniline: Piezocharge constants and electric potential study. Materials Letters, 220, 257-260.
  4. Bugnicourt, E., Cinelli, P., Lazzeri, A., Alvarez, V. A. (2014). Polyhydroxyalkanoate (PHA): Review of synthesis, characteristics, processing and potential applications in packaging.
  5. Bonartsev, A. P., Bonartseva, G. A., Reshetov, I. V., Kirpichnikov, M. P., Shaitan, K. V. (2019). Application of polyhydroxyalkanoates in medicine and the biological activity of natural poly (3-hydroxybutyrate). Acta Naturae, 11(2 (41)).

Reviewer 2 Report

The manuscript “Review of hybrid materials based on polyhydroxyalkanoates for biomedical applications” deals with the production of hybrid polyhydroxyalkanoate-based biomaterials with improved physico-mechanical, chemical, and piezoelectric properties and controlled biodegradation rates, for applications in bone, cartilage, nerve and skin tissue engineering.

A systematic study of the literature has been performed concerning the production techniques, material’s properties and applications. Several figures have been also added to represent some results in the field.

This is a very good and well organized work that covers all aspects from the production of bio-scaffolds to their applications. Therefore, the publication is recommended.

Only some minor revisions are required:

- Please, define acronyms the first time they appear, even if an Abbreviations section has been added at the end of the work.

- This review written by the research group of Prof. Maffulli can be added to the bone section, in order to enlarge the state of the art: Regeneration techniques for bone-To-Tendon and muscle-To-Tendon interfaces reconstruction, British Medical Bulletin, 2016, 117(1), pp. 25–37.

Author Response

  1. Please, define acronyms the first time they appear, even if an Abbreviations section has been added at the end of the work.

Answer: Thank you very much for this valuable comment. We have added acronyms throughout the text.

  1. This review written by the research group of Prof. Maffulli can be added to the bone section, in order to enlarge the state of the art: Regeneration techniques for bone-To-Tendon and muscle-To-Tendon interfaces reconstruction, British Medical Bulletin, 2016, 117(1), pp. 25–37.

Answer: Thank you very much for this valuable comment. We have added the reference to the paper as the reviewer recommended.

Round 2

Reviewer 1 Report

The authors have substantially revised the manuscript according to the comments.

The present form of the manuscript can be accepted for publication.